# OpenReview forum: "C$^3$AI: Crafting and Evaluating Constitutions for Constitutional AI"
_ACM.org/TheWebConf/2025/Conference — WWW 2025 Poster_

### Official Review · Reviewer_jG7Q · 2024-12-01

**Novelty:** 6
**Technical Quality:** 5

**Review:**

The paper focuses on "Constitutional AI", i.e. aligning LLMs with constitutions of their intended behavior. The paper presents the C3 framework that uses synthetic data augmentation and pairwise preference evaluation for finetuning and evaluating "constitutional" models.

The paper taps into an impressive array of principles gathered from various sources (Anthropic, Good for Humanity, Human Values, etc) and examines them with Llama-3-8B and various conversational evaluation datasets. Using Exploratory Graph Analysis, it illustrates how the principles correlate with each other, and how they cluster together. EGA is used to distill the large set of items into six main factors (F1-F6). The EGA method outperforms reasonable baselines.

Overall, the paper represents an interesting contribution to the development of "Constitutional AI" by presenting an approach that augments hand written constitutions with synthetic data that can then be used for fine tuning.

**Questions:**

I was left wondering to what extent item transformation and principle selection were evaluated directly, e.g. from samples of transformations. Do the transformations still convey the underlying principle? Do models make consistent decisions during principle selection? If these steps were not evaluated via human inspection, it remains unclear to what extent to model is trained correctly, even if the performance of the model in downstream tasks improves.

**Reviewer Confidence:**

4: The reviewer is certain that the evaluation is correct and very familiar with the relevant literature

**Scope:**

4: The work is relevant to the Web and to the track, and is of broad interest to the community

---

### Official Review · Reviewer_sZDJ · 2024-12-02

**Novelty:** 5
**Technical Quality:** 6

**Review:**

This paper introduces C$^3$AI framework which allows for creating effective constitutions for training safe large language models (llms). The paper build on the Anthropic's ideas of constitution, which includes a short list of principles that an LLM should consider while generating a response. The authors note that it is unclear to what extent a model adheres to the principles in a constitution and how they impact the model's behavior. The proposed C$^3$AI framework helps in identifying the most informative principles by constructing a constitution principle graph incorporating insights from AI and Psychologically.

I am not exactly sure how this relates to the Web. I understand that LLMs are ubiquitous and perhaps can be deployed to Web applications. However, the authors do not refer to any such applications. I think the authors need to justify how this benefits the Web community.

Section 5 was a bit unclear to me. Is it that you compare the constitution obtained through your EGA approach vis-à-vis what is deployed by Anthropic?

It was also a bit unclear how the graph in figure 2 was constructed.

Overall, I think this is a well-written paper and makes a significant contribution, only I am not sure if the Web Conference is the right venue.

**Questions:**

Could it be that a constitution obtained through your EGA approach have conflicting principles?

**Reviewer Confidence:**

3: The reviewer is confident but not certain that the evaluation is correct

**Scope:**

3: The work is somewhat relevant to the Web and to the track, and is of narrow interest to a sub-community

---

### Official Review · Reviewer_YyG5 · 2024-12-02

**Novelty:** 5
**Technical Quality:** 3

**Review:**

## Summary
The main content of the article is about the development of Constitutional AI (CAI), which is a framework for aligning advanced AI systems with diverse human values. The article discusses the process of crafting and evaluating constitutions for CAI models, including selecting and transforming items into principles, and selecting principles to form constitutions. It also presents a case study on aligning models with safety-related principles and demonstrates the effectiveness of the proposed framework.


## Strengths
1. The automated selection and filtering of suitable constitutions for RLAIF is an important and intriguing research problem.
2. The method proposed in the paper is easy to understand.

## Weaknesses
1. The main concern remains with the validation of the method. Most experiments in the paper are based on specific cases or rely on human intuition to understand the selection of principles. Regarding the effectiveness of the CAI model, the authors chose to validate it using OPRO on a LLaMA-3-8B model. On one hand, the variety and scale of the models used for validation are limited, and on the other hand, the RLHF methods are also restricted.

**Questions:**

1. For the selection of principles in RLAIF, not only is the choice of principles important, but the order of principles and the construction of the prompt also significantly impact the model's scoring. How did the authors determine the final prompt?
2. Given that the work focuses only on the selection of principles, specifically the reward model component, have the authors considered testing the effectiveness of the selected principles compared to the baseline on the Reward Bench [1]?

[1] RewardBench: Evaluating Reward Models for Language Modeling

**Reviewer Confidence:**

2: The reviewer is willing to defend the evaluation, but it is likely that the reviewer did not understand parts of the paper

**Scope:**

4: The work is relevant to the Web and to the track, and is of broad interest to the community

---

### Official Review · Reviewer_WTXd · 2024-12-02

**Novelty:** 6
**Technical Quality:** 5

**Review:**

In this work, the authors propose a framework called C3AI for developing constitutional AI models. Through through Exploratory Graph Analysis using human preference data on multiple factors like helpfulness, harmlessness, etc. they identify a subset of informative factors. They also find that agreement among humans regarding specific principles varies across factors. Lastly, they evaluate the impact of phrasing principles in positive vs negative manner,

Strengths:
1. The paper is well-written and descibes details of the framework, from item selection to model testing, well.
2. The idea of using principles from psychometric testing and evaluation in developing the constitutional AI frameworks is interesting (i.e, use of techniques like EGA)
3. Insights related to variation in principle-agreement, framing, safety-alignment related performance wins in comparison to baseline in the alignment finetuning step, etc.

Weaknesses:
1. It is confusing why the authors manually curated a set of 50 examples: how is this used in addition to the other human preference datasets used by the authors?
2. The literature review on the psychometric testing side is limited, and could further strengthen some of the analytical choices: for example, why was EGA chosen?
3. The performance improvement on using EGA-based measures for finetuning is small, and close to random chance. Can authors justify/hypothesize why this is the case, and improvements are only seen for safety-related principles?

**Questions:**

1. How have authors used the manually curated human preference dataset, in addition to the other human preference datasets used by the authors?
2. The performance improvement on using EGA-based measures for finetuning is small, and close to random chance. Can authors justify/hypothesize why this is the case, and improvements are only seen for safety-related principles?

**Reviewer Confidence:**

3: The reviewer is confident but not certain that the evaluation is correct

**Scope:**

3: The work is somewhat relevant to the Web and to the track, and is of narrow interest to a sub-community